# Stress and corticotropin releasing factor (CRF) promote necrotizing enterocolitis in a formula-fed neonatal rat model

Robert L. Bell[1,2,3], Ginger S. Withers[4], Frans A. Kuypers[2,5], Wolfgang Stehr[2,5¤‡]*, Aditi Bhargava [6‡]*

**1** East Bay Surgery Program, Department of Surgery, University of California San Francisco (UCSF) Benioff Children's Hospital, Oakland, California, United States of America, **2** Children's Hospital Oakland Research Institute, Oakland, California, United States of America, **3** The Permanente Medical Group, Department of Surgery, Walnut Creek, California, United States of America, **4** Department of Biology, Whitman College, Walla Walla, Washington, United States of America, **5** UCSF Benioff Children's Hospital Oakland, Oakland, California, United States of America, **6** Department of Obstetrics and Gynecology, Center for Reproductive Sciences, University of California San Francisco, San Francisco, California, United States of America

¤ Current address: Presbyterian Health System, Albuquerque, New Mexico, United States of America
‡ Co-senior authors.
* Aditi.bhargava@ucsf.edu (AB); wolfgangstehr@gmail.com (WS)

**Data Availability Statement:** All relevant data are within the manuscript.

**Funding:** This work was supported by funds from a R01 DK070878 from NIH/NIDDK to Aditi Bhargava.

## Abstract

The etiology of necrotizing enterocolitis (NEC) is not known. Alterations in gut microbiome, mucosal barrier function, immune cell activation, and blood flow are characterized events in its development, with stress as a contributing factor. The hormone corticotropin-releasing factor (CRF) is a key mediator of stress responses and influences these aforementioned processes. CRF signaling is modulated by NEC's main risk factors of prematurity and formula feeding. Using an established neonatal rat model of NEC, we tested hypotheses that: (i) increased CRF levels—as seen during stress—promote NEC in formula-fed (FF) newborn rats, and (ii) antagonism of CRF action ameliorates NEC. Newborn pups were formula-fed to initiate gut inflammation and randomized to: no stress, no stress with subcutaneous CRF administration, stress (acute hypoxia followed by cold exposure—NEC model), or stress after pretreatment with the CRF peptide antagonist Astressin. Dam-fed unstressed and stressed littermates served as controls. NEC incidence and severity in the terminal ileum were determined using a histologic scoring system. Changes in CRF, CRF receptor (CRFRs), and toll-like receptor 4 (TLR4) expression levels were determined by immunofluorescence and immunoblotting, respectively. Stress exposure in FF neonates resulted in 40.0% NEC incidence, whereas exogenous CRF administration resulted in 51.7% NEC incidence compared to 8.7% in FF non-stressed neonates (p<0.001). Astressin prevented development of NEC in FF-stressed neonates (7.7% vs. 40.0%; p = 0.003). CRF and CRFR immunoreactivity increased in the ileum of neonates with NEC compared to dam-fed controls or FF unstressed pups. Immunoblotting confirmed increased TLR4 protein levels in FF stressed (NEC model) animals vs. controls, and Astressin treatment restored TLR4 to control levels. Peripheral CRF may serve as specific pharmacologic target for the prevention and treatment of NEC.

A portion of Dr. Bhargava's salary was supported by this grant. The funders had no role in study design, data collection and analysis, decision to publish, or preparation of the manuscript. Other authors received no specific funding for this project.

**Competing interests:** No competing interests exist.

## Introduction

Necrotizing enterocolitis (NEC) is the most common fatal gastrointestinal (GI) disease affecting premature infants in the developed world [1]. The incidence is 0.3–2.4 cases of NEC for every 1,000 live births [2], corresponding to annual costs ranging between $500 million to $1 billion in the United States [3]. No specific therapy is available to treat NEC. Nearly one half of patients afflicted with NEC develop complications requiring surgical intervention; of these, approximately 50% die [1]. Overall mortality has remained unchanged over the past 30 years [3]. Survivors face ongoing morbidity due to malnutrition, recurrent small bowel obstructions, liver failure, and neurocognitive deficits [1, 4].

The major processes implicated in NEC's pathogenesis include abnormal bacterial colonization [5, 6], intestinal barrier dysfunction [4, 7–12], overzealous inflammation [11–16], and ischemia due to vasoconstriction [17–20]. While these have been well-characterized, their temporal and cause-effect relationships during NEC's development remain undefined. Widely accepted risk factors for NEC include prematurity, history of enteral formula feeding [2], and physiologic stress [4, 7]. Protective factors include breast-feeding [4], administration of probiotics [21], and corticosteroid administration [4, 22–24].

The peptide hormone corticotropin-releasing factor (CRF), the related urocortin (UCN) peptides, and their cognate CRF receptors (CRFRs) may play important roles in NEC's development. In mammals, CRF synthesis and secretion from the hypothalamus into the portal circulation initiates the response to physiologic and psychologic stress as a part of the hypothalamic-pituitary-adrenal (HPA) axis [25]. CRFRs ($CRF_1$ and $CRF_2$) are expressed ubiquitously in several cell types and organs [25], and are secreted into the plasma in extracellular vesicles [26]. $CRF_1$ is predominantly found in the brain, and its activation by CRF initiates the HPA axis response. $CRF_2$ is predominantly present in the periphery, and its activation by UCN1-3 returns stress responses back to homeostasis by facilitating negative feedback of the HPA axis [27, 28]. Through autocrine, paracrine, and endocrine mechanisms, CRF and urocortins act via CRFRs to elicit peripheral organ effects. Spatio-temporal activation of CRFRs and their ligands is nuanced and critical for disease development and progression [29]. Activation of $CRF_1$ is associated with pro-inflammatory events, whereas activation of $CRF_2$ is associated with anti-inflammatory effects in the GI tract as well as in mast cells [30].

Several studies have demonstrated that components of the CRF system modulate GI motility, barrier function, and inflammation [31–36]; these events also contribute to NEC's pathogenesis.

Luminal bacteria are necessary for NEC to occur [4]. CRF is associated with alterations in luminal bacterial colonization. Increased levels of endogenous CRF and its exogenous administration are associated with inhibited small bowel peristalsis [37, 38], altered secretion of luminal mucin [39] and gastric acid [40], and increased bacterial adherence to epithelial surfaces [39]. These result in bacterial overgrowth, loss of commensal bacterial species, and selection of pathogenic gram-negative and gas-forming bacterial organisms in the gut lumen [39, 41–43]. These luminal defenses have been found deficient in NEC [7, 12, 44], and similar shifts in the luminal microbiome are characteristic of NEC, in both experimental [5] and clinical [45] settings. In addition, CRF increases gut barrier permeability via increased expression of toll-like receptor 4 (TLR4) on enterocyte and immunocyte membranes [46, 47]. Downstream effects of TLR4 activation by bacterial endotoxin have been well characterized in NEC [1, 15], and include elaboration of pro-inflammatory cytokines [6], compromise of epithelial tight junctions [2], enterocyte apoptosis [9], and inhibition of enterocyte migration and restitution [48]. Independent of its TLR4-related actions, CRF serves as a chemoattractant and activator of mast cells and other immunocytes. Reciprocal release of nerve growth factor (NGF) from

immunocytes promotes innervation of these cells by the enteric nervous system (ENS), such that subsequent stress-induced CRF signaling by the ENS sustains a pro-inflammatory state [49–53]. CRF's actions in mast cells are mediated largely by $CRF_2$ [30]. Finally, CRF contributes to local vasoconstriction and enterocyte ischemia by promoting endothelin release, along with decreased endothelial nitric oxide synthase (eNOS) activity [54, 55]. Ischemic changes seen in NEC were traditionally attributed to asphyxia and hypoxic events [56]; however, more recent work suggests that NEC's ischemic insults stem from altered local endothelin-to-nitric oxide ratios [17–19] and endothelin receptor expression [20] favoring vasoconstriction, with resultant ischemia-reperfusion injury.

In addition, clinical factors that affect NEC's incidence are key modulators of CRF signaling. CRF signaling decreases with administration of probiotics [39, 41, 43], and CRF activity is subject to negative feedback control by corticosteroids [57]. In contrast, intestinal CRF activity increases in response to maternal separation and the transition from breast feeding to formula feeding [35, 58, 59]. Endogenous CRF and CRFR levels increase within the GI tract in response to inflammation [31] and stressful stimuli [52]. Newborn animals appear to be more vulnerable to these changes [57, 60]. This vulnerability may be accentuated in the setting of prematurity due to compromised feedback control from an immature HPA axis [57].

Given the parallels between peripheral CRF pathways and what we know about NEC, it is attractive to postulate a key role for CRF in NEC's pathogenesis. Peptide-based CRF inhibitors do not cross the blood-brain barrier and tend to have negligible effects on either central nervous system or HPA axis function. They do not appear to affect normal GI function. Thus, pharmacologic inhibition of overstimulated peripheral CRF signaling seems to offer promise for the prevention and treatment of NEC. In this proof-of-concept study, we sought to test the hypotheses that: (i) increased CRF levels—as seen during stress—promote NEC in formula-fed newborn rats, and (ii) antagonism of CRF action ameliorates NEC. We utilized a well-described neonatal rat model consisting of formula feeding and exposure to hypoxia and cold stress.

## Materials and methods

### Materials

CRF peptide and peptide antagonist, Astressin (AST) were purchased from American Peptide Company, Sunnyvale, CA.

### Animals

Animal experiments were performed using neonatal Sprague Dawley rats (Charles River, Pontage, MI) and were approved by the Institutional Animal Care and Use Committee of Children's Hospital Oakland Research Institute. Neonates were delivered spontaneously from timed-pregnant female rats. Rats were housed in temperature- and humidity-controlled cages that utilized a laminar flow HEPA filter system, unless specified. They had access to *ad lib* food, water, and environmental enrichment. Adult rats and neonates housed with dams were monitored daily. Neonates in experimental conditions were housed with members of their treatment groups in a temperature (35˚C) and humidity (15%)-controlled incubator. Each neonate was monitored, fed, and provided with bladder and bowel stimulation six times daily. All experiments were performed in accordance with the NIH and ARRIVE guidelines.

**Experimental groups.** Neonates (weight 6–10 g) were randomized into six treatment and two control groups on postnatal day 3: Group 1: dam-fed unstressed (DF, n = 22); Group 2: dam-fed stressed (DFS, n = 26); Group 3: formula-fed, unstressed (FF, n = 23); Group 4: formula-fed stressed (NEC, n = 25); Group 5: formula-fed unstressed with 30μg/kg CRF

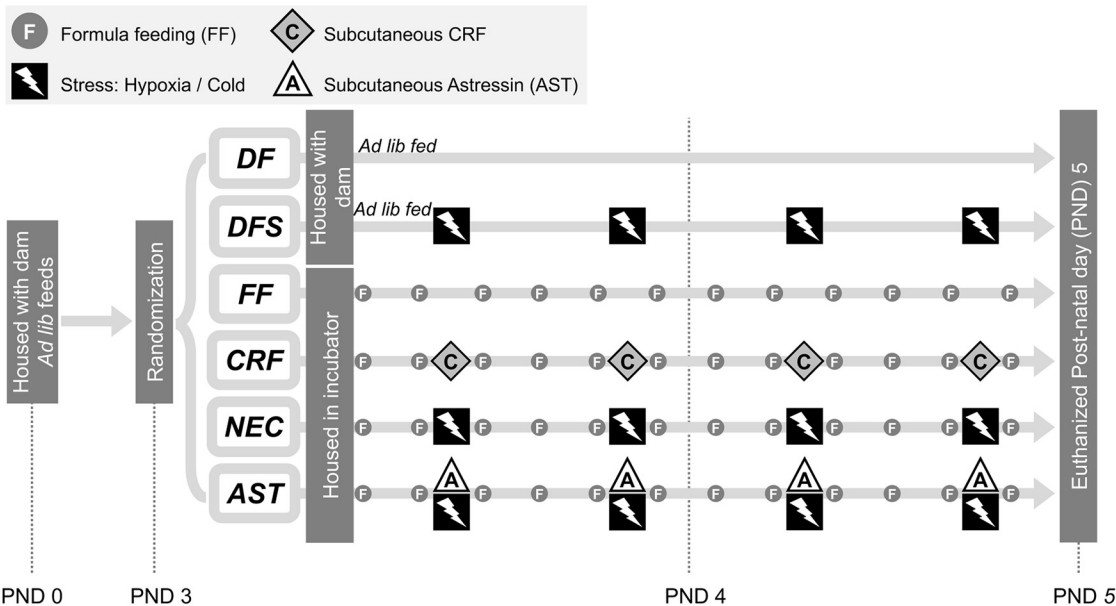

**Fig 1. Experimental model of NEC- a timeline.** Neonates were housed with dams for 3 days before being randomized at postnatal day (PND) 3 into 6 groups as shown. Dam-fed controls were randomized to no stress (DF) or hypoxia / cold stress exposure (DFS), housed with a dam, and allowed *ad libitum* feeding. Formula-fed neonates were separated from their dams and housed in an incubator. They were randomized to no stress (FF), hypoxia / cold stress exposure (NEC), no stress with CRF administration (CRF), and hypoxia / cold stress exposure after pretreatment with Astressin (AST).

administration (CRF, n = 27); Group 6: formula-fed stressed with 60μg/kg Astressin administration (AST, n = 26). Neonates in Groups 1 and 2 were housed with a dam and allowed *ad libitum* nursing (Fig 1).

## Experimental model of necrotizing enterocolitis (NEC)

Experimental NEC was induced as described previously, with a protocol that consisted of maternal separation, formula feeding, and exposure to hypoxia and cold stress [6, 61, 62]. Briefly, groups of formula-fed neonates (FF, NEC, CRF, and AST group) were separated from dams on postnatal day 3 and housed with other members of their treatment group in a temperature-controlled incubator (34°C). They were gavage fed six times daily with approximately 12.5μL/g body weight (80–200μL) of a special rodent formula consisting of 15g Similac 60/40 in 75 mL Esbilac canine milk replacement (Ross Pediatrics, Columbus, OH and Pet-Ag, Hampshire, IL, respectively). Stress sessions (DFS, NEC, and AST groups) took place twice daily, and consisted of exposure to a 100% $N_2$ atmosphere (hypoxia) in a modular incubation chamber (Billups-Rothenberg, Del Mar, CA) for 60 seconds, followed by exposure to 4°C for 10 minutes (cold stress). Animals were returned to their incubators immediately after stress sessions (Fig 1). CRF (30μg/kg) and Astressin (60μg/kg) in 100μL sterile water were administered twice-daily as subcutaneous injections. Astressin injections were performed 30 minutes before exposure to hypoxia and cold stress. Experimental conditions were applied for 48 hours, after which animals were euthanized as per the AVMA guidelines and specimens collected. Carbon dioxide exposure followed by bilateral thoracotomy and cardiac venting was used for euthanizing adult rats. Neonates were anesthetized with isoflurane and euthanized via decapitation. Experimental endpoints included body conditioning score of 2 or less, rectal prolapse, loss of vigorous mobility, lethargy, failure to respond to stimuli; as well as findings

suggestive of injury or aspiration that occurred during gavage feeding. These included respiratory distress, regurgitation of feeds, and pharyngeal bleeding. No rats died before meeting criteria for euthanasia.

## Histology

The GI tract was removed intact from euthanized pups, linearized, and gently flushed with 1mL of sterile phosphate-buffered saline (10mM PBS, pH 7.4). 40mm of terminal ileum was removed. The distal 20mm was flushed with fresh fixative (4% paraformaldehyde in PBS with 5% sucrose) and immersed in room-temperature fixative for 6 hours. The proximal 20mm was snap-frozen in liquid nitrogen and stored at -80°C for subsequent analysis (see below). Fixed specimens were rinsed and dehydrated in serial dilutions of ethanol in PBS (5, 10, 20, 50, and 70% ethanol), processed, embedded in paraffin, and sectioned at 5μm for microscopic analysis by the Mouse Pathology Core of the Helen Diller Cancer Center at the University of California at San Francisco. Sections were stained with hematoxylin and eosin (H&E) for light microscopy analysis. Paraffin-embedded slides were stored at room temperature for further analysis.

## Analysis of mucosal injury

Mucosal injury and presence of NEC were assessed using 5μm H&E-stained sections of intestine by researchers blinded to the treatment groups. Pathologic changes in intestinal architecture were evaluated via a NEC scoring system developed for use in neonatal rats [61, 62]. Histologic changes in ileum were scored on a scale of 0–3; 0 = normal, 1 = mild inflammation, separation of the villous core without other abnormalities, 2 = moderate inflammation, villous core separation, submucosal edema, and epithelial sloughing, and 3 = severe, denudation of epithelium with loss of villi, full-thickness necrosis, or perforation. Animals with histologic scores $\geq$ 2 were defined as having developed NEC (Fig 2).

## Antibodies

**Primary antibodies.** The primary and secondary antibodies, dilutions used, and sources were as follows: Antibodies from Santa Cruz Biotechnology, Santa Cruz, CA: $CRFR_{1/2}$ (sc-1757; goat; 1:500) [63], β-actin (A2228; mouse; 1:5,000) [29], TLR4 (M16) (sc-12511; goat; 1:1000), and CRF (rabbit; 1:5,000; Courtesy of Prof. W. Vale:) were used.

**Secondary antibodies.** For immunofluorescence staining goat anti-rabbit conjugated to Rhodamine Red-X or FITC (Jackson ImmunoResearch) at 1:500 dilution was used. For Western blot analyses donkey anti-goat/rabbit conjugated to Alexa Fluor 680 (Thermo Fisher Scientific) and donkey anti-mouse conjugated to IRDye 800 (Rockland Immunochemicals, Pottstown, PA) at 1:20,000 dilution was used.

## Western blot analysis

Ileum tissue samples were homogenized in RIPA buffer supplemented with protease inhibitor cocktail (Roche, Mannheim, Germany) and phosphatase inhibitor cocktails (Sigma-Aldrich). Total protein concentration was determined using the Bradford assay with bicinchoninic acid (BCA) reagent (Millipore Sigma, St. Louis, MO). Total protein (40 μg) was resolved by 10% SDS-PAGE, transferred to polyvinylidene difluoride membranes (PVDF, Immobilon-FL; Millipore, Billerica, MA) and blocked for 1 h at room temperature in Odyssey Blocking Buffer (Li-COR Biosciences, Lincoln, NE). Membranes were incubated with primary antibodies overnight at 4°C. Membranes were washed for 30 min (1 × PBS, 0.1% Tween20) and incubated

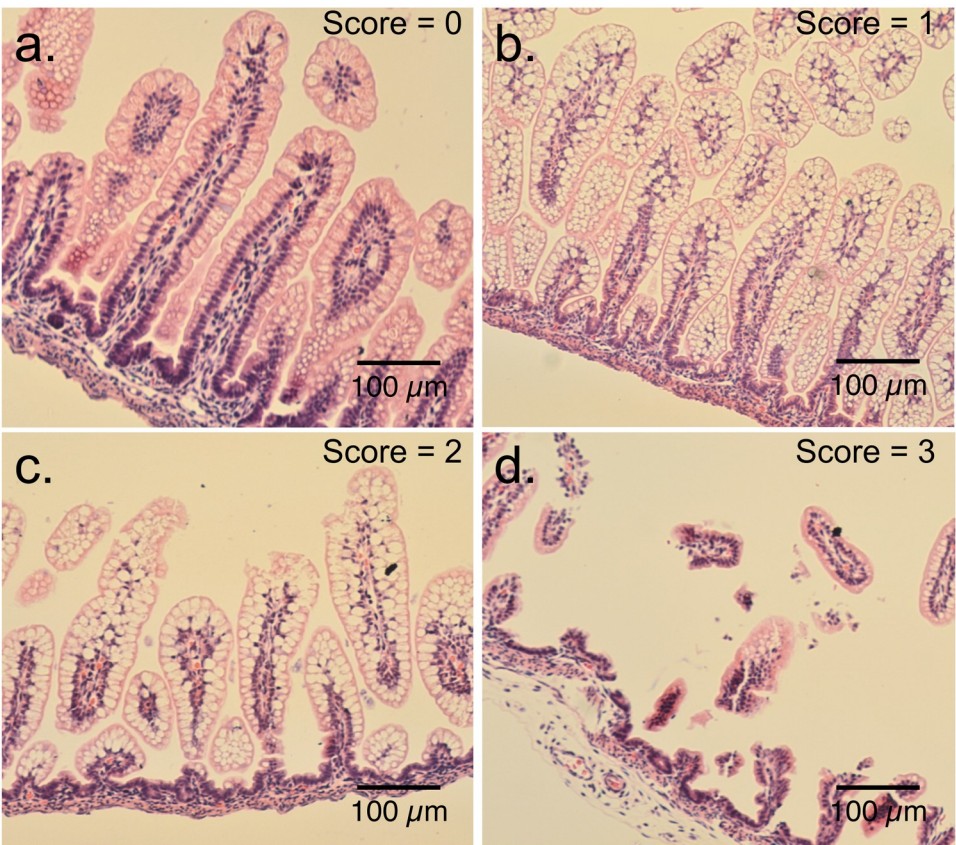

**Fig 2. Mucosal injury and NEC scoring system.** Representative H&E-stained terminal ileum sections to show mucosal injury consistent with NEC. (a) Normal intestinal mucosa: score 0. (b) Mild inflammation: score 1. (c) Moderate inflammation consistent with NEC: score 2. (d) Severe inflammation consistent with NEC: score 3.

with secondary antibodies for 1 h at room temperature. Blots were analyzed and quantified with the Odyssey Infrared Imaging System.

### Immunofluoroscopy and microscopy

Terminal ileum sections (5µM thick) from experimental and control groups were deparaffinized in xylene and rehydrated in ethanol series. Sections were incubated in blocking buffer containing 1x PBS, 0.3% Triton X-100, 10% normal goat serum for 1 h at room temperature followed by incubated with primary antibodies (anti-CRF and anti-CRF$_{1/2}$) overnight at 4˚C. Sections were washed and incubated with fluorescent secondary antibodies (conjugated to Rhodamine Red-X and FITC) for 1 h at room temperature. Images were acquired using an epi-fluorescence microscope (20x and 40x objectives) and images were captured using AxioVision Imaging software.

### Statistical analysis

The incidence of NEC was determined for each treatment group and expressed as percentage ± standard error. Groups were compared using directional Chi square analysis and Fischer's exact test using Stata SE software (StatCorp; College Station, TX). $P$ values $< 0.05$ were considered statistically significant. All data are representative of at least three independent experiments (biological replicates), involving 12 litters of neonatal rats.

## Results

### Formula feeding by itself causes mild inflammation in neonates

Stress in dam-fed neonates or formula feeding in absence of other factors may not be sufficient to cause overt NEC-like disease in rodents or humans. To test this notion, neonates were left in their home cages with dams without any handling or stress (dam-fed; DF) or exposed to acute hypoxia and cold stress (DFS). As expected, terminal ileum histology was normal in DF pups and after 72 hours of exposure to stressors, none (0/26) of the pups in DFS group developed histologic findings consistent with NEC (Fig 3a, 3b and 3g). Formula feeding without stress exposure (FF) resulted in mild inflammatory changes in the terminal ileum with 17 of 23 pups demonstrating mild inflammation with vacuolization of villi and two of 23 (8.7 ± 5.9%) developed NEC (Fig 3c and 3g).

### Formula feeding combined with hypoxia and cold stress exposure causes NEC-like changes in the gut morphology

We confirmed that formula feeding combined with acute exposure to hypoxia and cold stress (NEC) over 48 hours caused overt changes in gut histopathology. Moderate to severe inflammatory changes occurred in 10 of 25 pups (40.0 ± 9.8%) animals in the NEC group, with nearly all specimens demonstrating some degree of inflammatory change (23 of 25 pups; Fig 3d and 3g). Histopathological changes were also accompanied by gross changes in the small bowel that showed erythema, edema, full thickness necrosis, and perforation in pups with NEC compared with DF unstressed controls (Fig 4a). Gross changes observed in rat gut were similar to those seen in preterm newborn human infants with complicated NEC (Fig 4b).

### CRF and CRF receptor expression is increased after induction of NEC

We next asked if NEC is associated with increased expression of CRF and its receptors in the terminal ileum. Immunofluorescence staining revealed diffuse and low levels of CRF and CRFR immunoreactivity (CRF-IR and $CRF_{1/2}$-IR, respectively) in the dam-fed unstressed group (Fig 5a; DF). Exposure to hypoxia and cold stress in dam-fed animals increased CRF-IR in the villi (Fig 5b; DFS). CRF and CRFRs co-localized in the ileum (Fig 5, Merge). In the formula-fed unstressed group, again CRF-IR co-localized with its receptors along the basolateral aspects of villous enterocytes, along with some staining within villi and in the submucosal and myenteric plexuses (Fig 5c; FF). Induction of NEC resulted in clear, discrete and multiple points of co-localization of CRF-IR with its receptors within submucosal and myenteric plexuses, and in the villi (Fig 5d; NEC), and omission of primary antibody (negative control) did not show any signal (Fig 5e). Analysis of the sections at higher magnification revealed CRF-IR and $CRF_{1/2}$-IR co-localization in the villus tip with little to no staining in the neurons of the submucosal or myenteric plexuses in DF control neonates. In NEC neonates, CRF-IR and $CRF_{1/2}$-IR expression increased in the villi and around the center corresponding to the location of enteric neurons coursing alongside blood vessels, and was diffused and disorganized. Expression was also clearly evident in the neurons of the plexuses (Fig 5f). This finding suggests an association between the development of NEC and increased CRF and CRFR expression within the enteric nervous system as well as the enterocytes.

### Exogenous CRF promotes NEC-like changes in the gut morphology

Having confirmed our hypothesis that hypoxia and cold stress exposure (NEC group) results in increased levels of CRF, we asked if CRF alone is sufficient to initiate NEC-like disease. We administered CRF in formula-fed unstressed rats instead of hypoxia and cold stress exposure

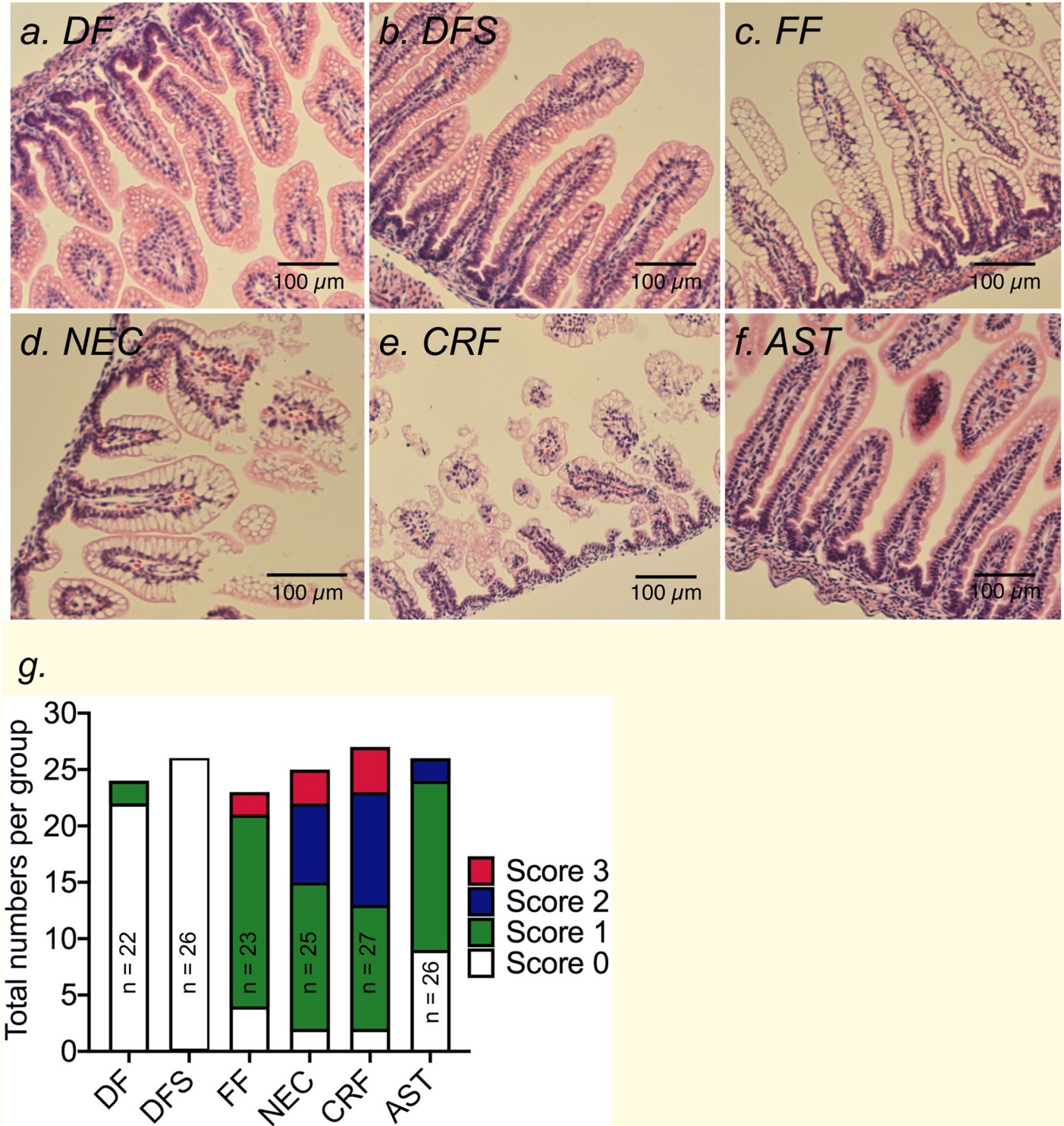

**Fig 3. Formula feeding and exposure to acute stressors causes frank NEC-like histopathological damage in rats.** Representative H&E-stained micrographs showing villi damage in NEC. (a-c) Terminal ileum sections of DF, DFS, and FF unstressed pups showed normal gut histology with well-preserved villi structure. (d) Inflammatory changes were present in terminal ileum of FF neonates exposed to stressors (NEC) or (e) FF unstressed pups with CRF administration. (f) Pretreatment with Astressin in FF pups prevented stress-induced changes in ileum histopathology and prevented development of NEC. DF = dam-fed, unstressed; DFS = dam fed, stressed; FF = formula-fed unstressed; NEC = formula-fed, stressed; CRF = formula-fed, unstressed with 30µg/kg of sc CRF administration; AST = formula-fed, stressed with 60µg/kg of sc CRF antagonist, Astressin administration. Scale Bar = 100µM. (g) Stack bar graph summarizing numbers of neonates with 0–3 scores within control or treatment groups.

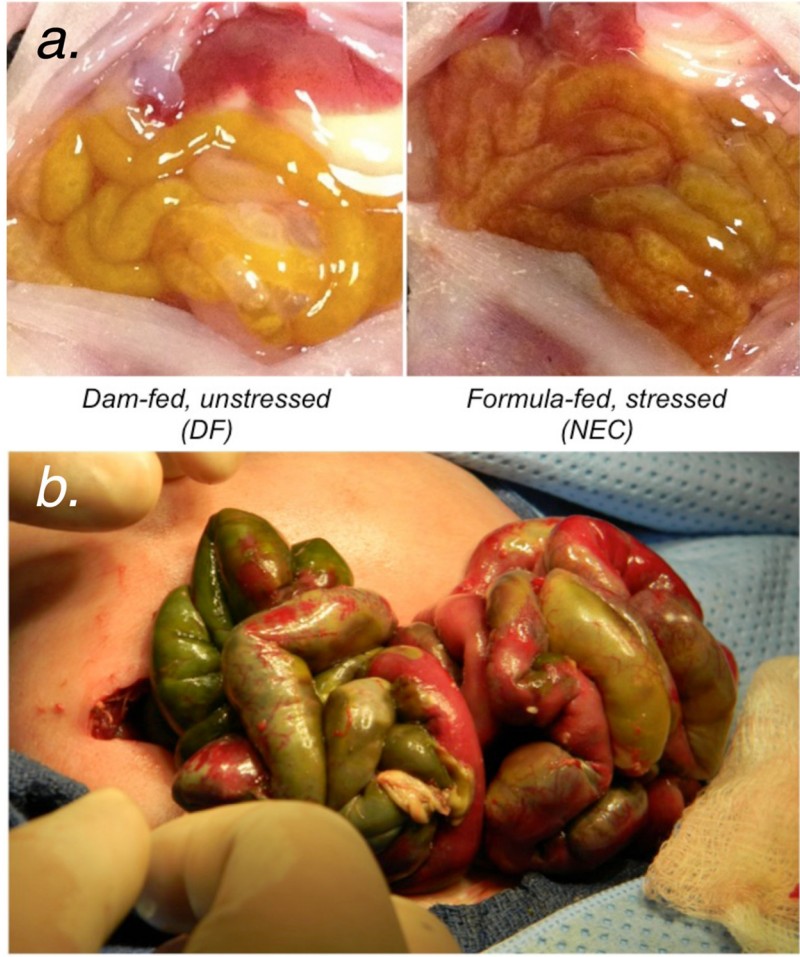

**Fig 4. Experimental model of necrotizing enterocolitis.** (a) Formula feeding combined with hypoxia and cold stress exposure induced gross changes in the small bowel including erythema, edema, and full thickness necrosis. (b) Gross intra-operative findings representative of complicated NEC in a premature human neonate.

(CRF group). As predicted, 51.9 ± 9.6% (14 of 27) of neonates developed NEC (Fig 3a and 3b). This increase in incidence reached statistical significance compared to DF, DFS, and FF groups (p < 0.001, Fig 6).

### CRF antagonism with Astressin prevents development of NEC-like changes in the gut morphology

We reasoned if exogenous CRF was sufficient to promote NEC-like inflammation and gross gut edema, antagonism of CRF even in formula-fed neonates exposed to hypoxia and cold stress should ameliorate these changes. As predicted, AST administration abrogated development of NEC in 92% of the pups with 24 of 26 pups showing no or low-grade inflammation (score 0–1, Fig 3). AST treatment significantly reduced NEC incidence to 7.7 ± 5.2% (2 of 26) versus 51.9 ± 9.6% (14 of 27) in the CRF group and 40.0 ± 9.8% (10 of 25) in the NEC group (p = 0.0033; Fig 6).

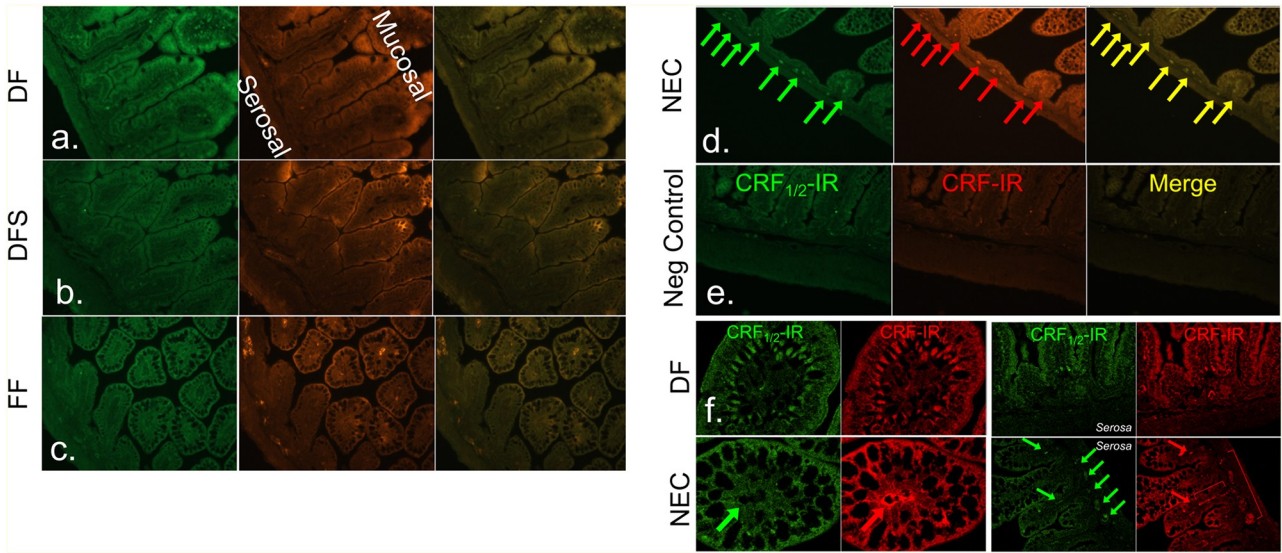

**Fig 5. CRF and CRF receptor immunoreactivity (IR) is increased in the terminal ileum of neonates with NEC.** (a-e) Representative immunostained section from DF, DFS, FF and NEC ileum. CRF-IR (red) and CRFR (CRF$_{1/2}$-IR, green) was evident in the villi, blood vessels, and neurons within the myenteric plexuses of FF and NEC groups, but only low, diffuse staining was seen in DF and DFS control groups. (f) Higher magnification (63x) confocal images revealed differences in staining pattern in CRF-IR and CRF$_{1/2}$-IR in DF controls versus NEC groups.

## Astressin treatment decreases toll-like receptor 4 (TLR4) levels

CRF is known to increase expression of TLR4, which in turn causes changes in gut permeability. Next, we confirmed using western blotting that TLR4 levels were increased in ileum of neonates with NEC compared with DF unstressed controls. Since treatment with Astressin prevented development of NEC, we ascertained TLR4 levels in ileum of neonates in AST group and found expression levels to be similar to those seen in DF unstressed control group (Fig 7). This data suggests that antagonism of CRF with Astressin was sufficient to downregulate TLR4 levels.

## Discussion

Necrotizing enterocolitis is a major cause of morbidity and mortality in premature neonates. Despite being first described over 100 years ago, no specific treatments have been developed. The role of stress in the development of NEC has been established. However, little is known about the role of key mediators of the stress axis—such as CRF and CRFRs—in NEC. In this study, we demonstrated that (i) stress in combination with formula-feeding, but neither alone, cause NEC-like histologic changes; (ii) over- and mis-expression of CRF is associated with the development of NEC; and (iii) CRF antagonism is sufficient to markedly decrease NEC incidence in an experimental animal model. While most studies that employed a similar NEC model, demonstrated NEC rates over 50% [6, 61, 62]; ours did not reach that level. This is likely explained by the fact that we limited our experimental conditions to 48 hours after randomization, whereas others applied their models over a 72-hour period.

Formula feeding is known to increase expression of the components of the CRF system, whereas administration of probiotics, breast milk, and corticosteroids decrease their expression [39, 41, 43, 57]. Human milk also has antioxidant properties; in experimental NEC, glutamine and arginine supplementation has beneficial effects due to alterations in lipid peroxidation and antioxidant enzyme levels in the small intestine [64, 65]. Formula milk and other

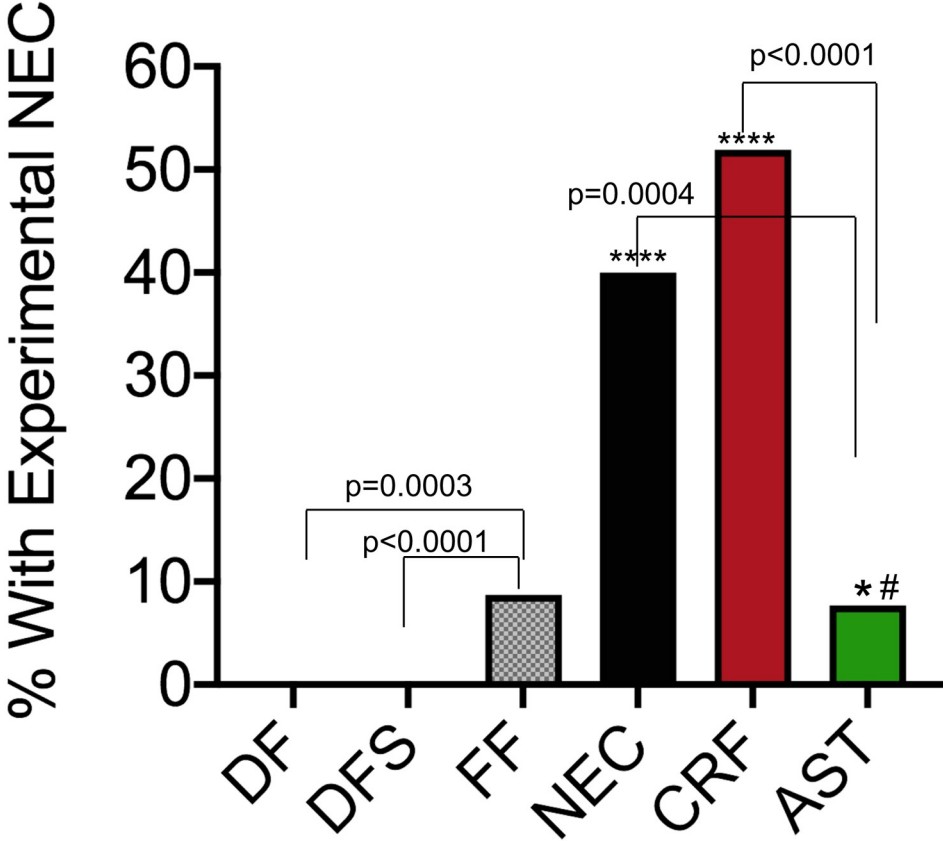

**Fig 6. Incidence of necrotizing enterocolitis among treatment groups.** The incidence of NEC was determined for each treatment group and expressed as percentage ± standard error. Groups were compared using directional Chi square analysis and Fischer's exact test. *: FF versus NEC groups, p = 0.006; **: FF versus CRF groups, p < 0.001; #: NEC versus AST groups, p = 0.033. DF = dam-fed, unstressed; DFS = dam fed, stressed; FF = formula-fed unstressed; NEC = formula-fed, stressed; CRF = formula-fed, unstressed with 30μg/kg of sc CRF administration; AST = formula-fed, stressed with 60μg/kg of sc CRF antagonist, Astressin administration.

chemicals also alter antioxidant levels, but whether AST treatment has beneficial effects on antioxidant enzymes [65, 66], is an area that needs further investigation. Spatio-temporal activation of CRFRs and their ligands is nuanced and critical for development of several GI disorders that include inflammatory bowel disease and functional GI diseases [29]. Stress-induced alterations in GI motility and diarrhea are well described, and gut-specific elimination of CRF ameliorates these symptoms. Activation of $CRF_1$ is associated with pro-inflammatory events, whereas activation of $CRF_2$ is associated with anti-inflammatory effects in the GI as well as in mast cells [30]. Newborn animals appear to be more vulnerable to these changes [60]. In this study, we found increased expression of CRF in enteric neurons in the terminal ileum of rats with NEC. Co-localization of CRF and CRFR expression within submucosal and myenteric plexuses and also within the core of villi was more robust, albeit disorganized in rats with NEC, whereas organized basolateral localization was evident in the ileum of control rats. This is the first report to demonstrate upregulation of CRF immunoreactivity in the gut of rats with experimental NEC. Although previous literature has suggested that CRF activity might play a role in neonatal intestinal injury and repair [67], here, we demonstrate unequivocally that stress-induced increases in CRF are sufficient to increase NEC incidence and severity.

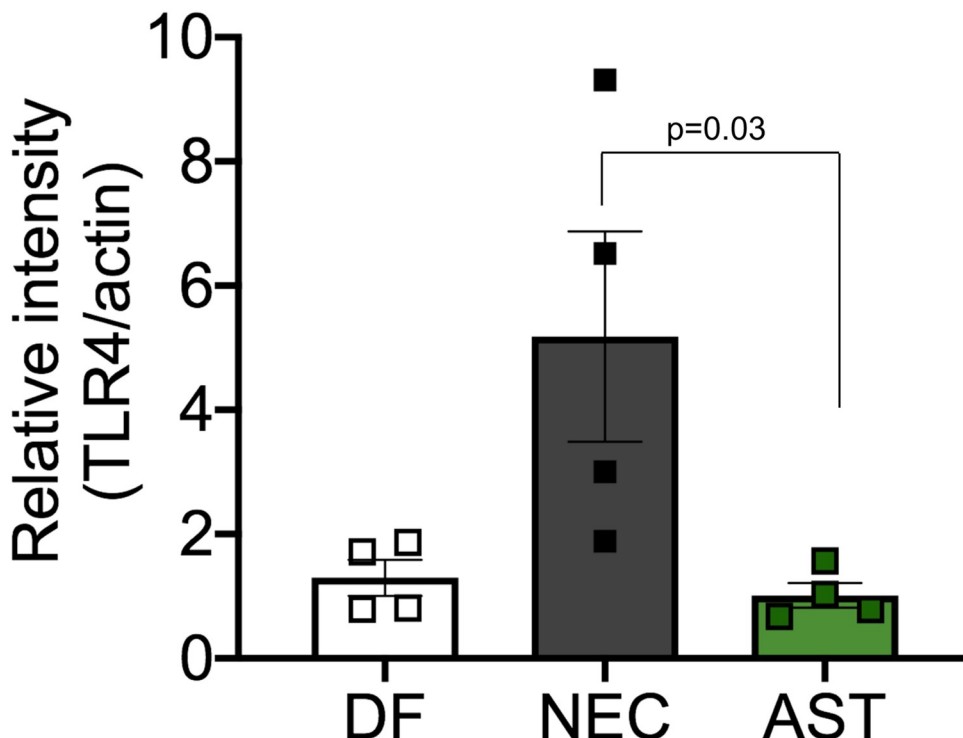

**Fig 7. TLR4 levels in NEC.** Bar graph showing increased TLR4 levels in rats with NEC and AST treatment decreased levels to those seen in dam-fed rats. Actin was used as normalization control. P = 0.03 NEC versus AST.

We further demonstrated that exogenous CRF administration promotes the development of NEC even in absence of external stressors, primarily via mucosa epithelial inflammation leading to villi loss, submucosal edema, necrosis, and perforation. In support of this pro-inflammatory role of CRF, pharmacological antagonism of CRF action was protective; we found NEC incidence was decreased by ~81% in rats after CRF antagonism even in the face of formula-feeding and exogenous stressors. CRF antagonism was accompanied by less severe mucosal injury compared to rats with NEC. Previous studies have described the role of CRF activation in various inflammatory gut disorders and in NEC's key pathologic processes [37–43, 46, 47, 49–51, 53–55]; here, we show that antagonizing the actions of CRF can prevent development of experimental NEC.

Alteration in intestinal barrier permeability is a hallmark of human NEC. TLR4 activation, bacterial overgrowth, and vasoconstriction are thought to promote gut ischemia and intestinal barrier permeability in NEC. Increased TLR4 expression and activation are key steps in NEC's development, and have been shown to precede overt histologic signs of inflammation in experimental NEC [6, 9, 61]. Similar to others, we found TLR4 expression was increased in ileum of rats with NEC compared with controls. CRF antagonism markedly decreased TLR4 expression. Other studies have shown contribution of CRF in modulating mast cells and gut function including motility and permeability [33, 34, 68, 69]. While this study did not ascertain the contribution of immune versus non-immune TLR4 in promoting NEC, CRFRs are present in both immune and non-immune cells of the gut. Both endocrine and paracrine actions of CRF have been described in these cell types [25].

Stress in neonates has been shown to be associated with mucosal injury in a variety of gut disorders. CRF and CRFR activation have been shown to be key modulators in the brain-gut

axis. This study demonstrates that CRF activation plays a role in the development of experimental NEC via increased receptor localization and disorganization leading to mucosal injury. These findings suggest that at least in the setting of experimental NEC, specific antagonism of CRF in the peripheral tissues ameliorates NEC's incidence and severity, and holds promise for pharmacologic prevention of this disease.

## Acknowledgments

We thank Min Liao in the Bhargava lab for technical help.

## Author Contributions

**Conceptualization:** Robert L. Bell, Frans A. Kuypers, Wolfgang Stehr, Aditi Bhargava.

**Data curation:** Robert L. Bell.

**Formal analysis:** Robert L. Bell, Ginger S. Withers, Aditi Bhargava.

**Funding acquisition:** Aditi Bhargava.

**Investigation:** Robert L. Bell, Ginger S. Withers, Frans A. Kuypers, Aditi Bhargava.

**Methodology:** Robert L. Bell, Ginger S. Withers, Frans A. Kuypers, Wolfgang Stehr, Aditi Bhargava.

**Project administration:** Frans A. Kuypers, Wolfgang Stehr, Aditi Bhargava.

**Resources:** Frans A. Kuypers, Wolfgang Stehr, Aditi Bhargava.

**Supervision:** Frans A. Kuypers, Wolfgang Stehr, Aditi Bhargava.

**Validation:** Robert L. Bell.

**Writing – original draft:** Robert L. Bell, Aditi Bhargava.

**Writing – review & editing:** Robert L. Bell, Ginger S. Withers, Frans A. Kuypers, Wolfgang Stehr, Aditi Bhargava.

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
