## [Decision Letter · Decision Letter 0]

22 Mar 2021

PONE-D-21-01479

Stress and corticotropin releasing factor (CRF) promote necrotizing enterocolitis in a formula-fed neonatal rat model

PLOS ONE

Dear Dr. Bhargava,

Thank you for submitting your manuscript to PLOS ONE. After careful consideration, we feel that it has merit but does not fully meet PLOS ONE’s publication criteria as it currently stands. Therefore, we invite you to submit a revised version of the manuscript that addresses the points raised during the review process.

The paper could have an important technical contribution in the area. I agree with all reviewers think that mainly the manuscript have several technical deficiencies. There are a lot of careless mistakes in the manuscript and the whole manuscript shall be reviewed by the authors carefully. The article requires some technical revisions and a thorough proof-read for grammatical errors.

We look forward to receiving your revised manuscript.

Kind regards,

Arda Yildirim, Ph.D.

Academic Editor

PLOS ONE

Journal Requirements:

3. At this time, we request that you  please report additional details in your Methods section regarding animal care, as per our editorial guidelines:

(a) Please provide details of animal welfare (e.g., shelter, food, water, environmental enrichment)

(b) Please include the method of euthanasia

(c) Please provide additional information about the experimental endpoints used in the model.

Please provide the frequency of monitoring and specific criteria used to determine when animals should be euthanized.

In addition, please state whether any animals died before meeting the criteria for euthanasia.

Thank you for your attention to these requests.

4. Please ensure you have discussed any potential limitations of your study in the Discussion.

'DK070878 to AB '

Please clarify, naming in full, the sources of funding (financial or material support) for your study. List the grants or organizations that supported your study, including funding received from your institution.State what role the funders took in the study. If the funders had no role in your study, please state: “The funders had no role in study design, data collection and analysis, decision to publish, or preparation of the manuscript.”If any authors received a salary from any of your funders, please state which authors and which funders.If you did not receive any funding for this study, please state: “The authors received no specific funding for this work.”

6. Please include a caption for figure 7.

Additional Editor Comments:

For your guidance, you can check the reviewers' comments. Thank you for giving us the opportunity to consider your work.

Reviewers' comments:

Reviewer's Responses to Questions

**Comments to the Author**

1. Is the manuscript technically sound, and do the data support the conclusions?

Reviewer #1: Yes

Reviewer #2: Yes

2. Has the statistical analysis been performed appropriately and rigorously? 

Reviewer #1: Yes

Reviewer #2: Yes

3. Have the authors made all data underlying the findings in their manuscript fully available?

Reviewer #1: Yes

Reviewer #2: Yes

4. Is the manuscript presented in an intelligible fashion and written in standard English?

Reviewer #1: Yes

Reviewer #2: Yes

5. Review Comments to the Author

Reviewer #1: The study on Stress and corticotropin releasing factor (CRF) promote necrotizing enterocolitis in a

formula-fed neonatal rat model was an interesting study with novel ideas and conclusions on the role of CRF showing promising pharmacological role in treatment or prevention of necrotising enterocolitis in the neonates. However, there are some questions and concerns that need clarification.

1. What determined the quantity of formula feed given to each neonates? Is the quantity given not excessive, thereby responsible for the observed inflammatory damages in the bowel of the neonates.

2. What other markers of oxidative stress did you determine in the course of the study and the probable roles of antioxidants instead of CRF on the oxidative stress and inflammation in the affected neonates.

There are specific observations in some of the sections that may require your correction.

Abstract

1. Line 50: Stressed neonates, how were they stressed. Please indicate in one or two words.

2. Lines 60-61: The conclusion was not explicit. Too shallow because much of the findings were not captured.

Introduction

Lines 136-138: The hypothesis mentioned in the justification for the study is not entirely the same as the ones mentioned in the abstract. Please adjust it!

Materials and methods

1. Line 195: Evaluation of specific oxidative stress markers from post mitochondrial fractions of homogenized intestinal tissue of ought to have been added. Like lipid peroxidation, glutathione levels and antioxidant enzymes such as superoxide dismutase.

2. Line 197: Names or initial of researchers are not necessary here, why not put them in the acknowledgment if they are not part of the authors, or authors' contributions.

3. Mention the parameters evaluated in the antibodies, Western blot and immunoflourescence analysis.

Results

1. Line 337: AST, render in full and put the abbreviation in bracket.

Discussion

1. Line 401-402: Please cite appropriate reference.

Reviewer #2: The manuscript entitled as 'Stress and corticotropin releasing factor (CRF) promote necrotizing enterocolitis in a

formula-fed neonatal rat model' is well designed l study. However, minor revision is needed, than it can be published in this journal. Suggestions are indicated on the manuscript.

6. PLOS authors have the option to publish the peer review history of their article (what does this mean?). If published, this will include your full peer review and any attached files.

Reviewer #1: **Yes: **Dr O. I Azeez

Reviewer #2: No

---

## [Author Response · Author response to Decision Letter 0]

29 Apr 2021

Arda Yildirim, Ph.D.

Academic Editor

PLOS ONE

RE: PONE-D-21-01479

Dear Dr. Yildirim,

Please find attached a revised version of our manuscript entitled “Stress and corticotropin releasing factor (CRF) promote necrotizing enterocolitis in a formula-fed neonatal rat model”. Changes in text are in blue. Below, we provide a point-by-point responses to reviewers.

Reviewer #1: 

The study on Stress and corticotropin releasing factor (CRF) promote necrotizing enterocolitis in a formula-fed neonatal rat model was an interesting study with novel ideas and conclusions on the role of CRF showing promising pharmacological role in treatment or prevention of necrotising enterocolitis in the neonates. However, there are some questions and concerns that need clarification.

1. What determined the quantity of formula feed given to each neonates? Is the quantity given not excessive, thereby responsible for the observed inflammatory damages in the bowel of the neonates.

Response. The quantity of formula feed was informed by established neonatal rat models of necrotizing enterocolitis and approved by our IACUC committee’s veterinarian. Overfeeding is a valid concern and was addressed during study design. It is notable that formula feeding—and perhaps overfeeding—is associated with the development of NEC in humans. In our experiments, formula feeding alone (i.e. without exposure to physiologic stress or corticotropin releasing factor) was associated with some inflammatory changes, but was not clearly associated with the development of overt NEC, thus formula-feeding alone is not associated with the observed effects. These changes were ameliorated with the administration of the CRF antagonist, Astressin.

2. What other markers of oxidative stress did you determine in the course of the study and the probable roles of antioxidants instead of CRF on the oxidative stress and inflammation in the affected neonates.

Response. We thank the reviewer for this interesting question. In this study, we did not ascertain the roles of antioxidants. Determining the roles of antioxidants will require exhaustive set of experiments which is beyond the scope of this paper. We further feel that addition of more data will distract from message and the role of CRF system in NEC. As suggested, this will be a great follow-up study. We have discussed the potential role of oxidative stress in our discussion.

There are specific observations in some of the sections that may require your correction.

Abstract

1. Line 50: Stressed neonates, how were they stressed. Please indicate in one or two words.

Response. Lines 48-49 already describe the stressor used (acute hypoxia followed by cold exposure).

2. Lines 60-61: The conclusion was not explicit. Too shallow because much of the findings were not captured.

Response. We are perplexed as conclusions are meant to be short and general. We feel that given the word limits for an abstract, our conclusion statement is appropriate. Our discussion contains more explicit and in-depth conclusion. 

Introduction

Lines 136-138: The hypothesis mentioned in the justification for the study is not entirely the same as the ones mentioned in the abstract. Please adjust it!

Response. We thank the reviewer for catching this discrepancy. We have modified the hypothesis in lines 136-138 to be congruent.

Materials and methods

1. Line 195: Evaluation of specific oxidative stress markers from post mitochondrial fractions of homogenized intestinal tissue of ought to have been added. Like lipid peroxidation, glutathione levels and antioxidant enzymes such as superoxide dismutase.

Response. We thank the reviewer for suggesting alternative mechanisms. Indeed, as the reviewer points out, preterm babies as well as NEC babies are exposed to more oxidative stress than full-term babies. We have added this to our discussion (Lines 383-387) and provided 3 new references (#s 64-66). However, the tease out which precise oxidative stress pathway intersects with the CRF system, will require exhaustive sets of new experiments along with use of additional pharmacological agents. This is beyond the scope of this study.

2. Line 197: Names or initial of researchers are not necessary here, why not put them in the acknowledgment if they are not part of the authors, or authors' contributions.

Response. Initials of the researchers help with further identifying their roles but as suggested, we have removed the initials from the methods.

3. Mention the parameters evaluated in the antibodies, Western blot and immunoflourescence analysis.

Response. All sources of antibodies and dilutions used are provided (Lines 224-231). We are unclear as to what other parameters are required?

Results

1. Line 337: AST, render in full and put the abbreviation in bracket.

Response. AST is first defined in line 144, under materials section.

Discussion

1. Line 401-402: Please cite appropriate reference.

Response. Reference has been provided.

Reviewer #2: 

The manuscript entitled as 'Stress and corticotropin releasing factor (CRF) promote necrotizing enterocolitis in a formula-fed neonatal rat model' is well designed l study. However, minor revision is needed, than it can be published in this journal. Suggestions are indicated on the manuscript.

1. Define FF (formula fed, line 46).

2. The companies or sources are provided in parenthesis and we have removed the sentences that created this confusion.

All suggestions indicated on the manuscript have been incorporated.

Editorial comments to be included in the manuscript: 

Details of animal welfare – shelter, food, water, environmental enrichment –

- Adult animals were housed in temperature- and humidity-controlled cages that utilized a laminar flow HEPA filter system. They had access to ad lib food, water, and environmental enrichment. Adult animals and neonates housed with dams were monitored at least daily. Neonates in experimental conditions were housed with members of their treatment groups in a temperature (35ºC) and humidity (15%)- controlled incubator. Each of these animals was inspected, fed, and provided with bladder and bowel stimulation six times daily. 

Method of euthanasia – 

Carbon dioxide exposure followed by bilateral thoracotomy and cardiac venting was used for euthanizing adult rats. Neonates were anesthetized with isoflurane and euthanized via decapitation.

Information about experimental endpoints used in the model

Besides gavage-induced thoracic injury, other experimental endpoints included body conditioning score of 2 or less and rectal prolapse.

Frequency of monitoring – at least once daily during the experiment

- Specific criteria used to determine which animals should be euthanized – 

o Criteria for euthanasia of neonates included loss of vigorous mobility, lethargy, failure to respond to stimuli; as well as findings suggestive of injury or aspiration that occurred during gavage feeding. These included respiratory distress, regurgitation of feeds, and pharyngeal bleeding.

- Any animals died before meeting criteria for euthanasia

o No

We hope that our manuscript is now acceptable for publication.

Sincerely,

Aditi Bhargava

---

## [Decision Letter · Decision Letter 1]

25 May 2021

Stress and corticotropin releasing factor (CRF) promote necrotizing enterocolitis in a formula-fed neonatal rat model

PONE-D-21-01479R1

Dear Dr. Bhargava,

We’re pleased to inform you that your manuscript has been judged scientifically suitable for publication and will be formally accepted for publication once it meets all outstanding technical requirements.

Kind regards,

Arda Yildirim, Ph.D.

Academic Editor

PLOS ONE

Additional Editor Comments (optional):

Thanks for your hard work.

Reviewers' comments:

Reviewer's Responses to Questions

**Comments to the Author**

1. If the authors have adequately addressed your comments raised in a previous round of review and you feel that this manuscript is now acceptable for publication, you may indicate that here to bypass the “Comments to the Author” section, enter your conflict of interest statement in the “Confidential to Editor” section, and submit your "Accept" recommendation.

Reviewer #1: All comments have been addressed

Reviewer #2: All comments have been addressed

2. Is the manuscript technically sound, and do the data support the conclusions?

Reviewer #1: Yes

Reviewer #2: Yes

3. Has the statistical analysis been performed appropriately and rigorously? 

Reviewer #1: Yes

Reviewer #2: Yes

4. Have the authors made all data underlying the findings in their manuscript fully available?

Reviewer #1: Yes

Reviewer #2: Yes

5. Is the manuscript presented in an intelligible fashion and written in standard English?

Reviewer #1: Yes

Reviewer #2: Yes

6. Review Comments to the Author

Reviewer #1: (No Response)

Reviewer #2: The manuscript is well designed and also all required suggestion/corrections performed. Now it is suitable to be publish in the journal.

7. PLOS authors have the option to publish the peer review history of their article (what does this mean?). If published, this will include your full peer review and any attached files.

Reviewer #1: No

Reviewer #2: No

---

## [Editor Report · Acceptance letter]

1 Jun 2021

PONE-D-21-01479R1 

Stress and corticotropin releasing factor (CRF) promote necrotizing enterocolitis in a formula-fed neonatal rat model 

Dear Dr. Bhargava:

I'm pleased to inform you that your manuscript has been deemed suitable for publication in PLOS ONE. Congratulations! Your manuscript is now with our production department. 

Kind regards, 

on behalf of

Prof. Dr. Arda Yildirim 

Academic Editor

PLOS ONE